# Electromagnetohydrodynamic Electroosmotic Flow and Entropy Generation of Third-Grade Fluids in a Parallel Microchannel

**DOI:** 10.3390/mi11040418

**Published:** 2020-04-16

**Authors:** Chunhong Yang, Yongjun Jian, Zhiyong Xie, Fengqin Li

**Affiliations:** School of Mathematical Science, Inner Mongolia University, Hohhot, Inner Mongolia 010021, China; Yangch@imu.edu.cn (C.Y.); xiezhiyong@mail.imu.edu.cn (Z.X.); lifq@imu.edu.cn (F.L.)

**Keywords:** third grade fluids, entropy generation, heat transfer, electromagnetohydrodynamic (EMHD) flow, electroosmotic flow (EOF)

## Abstract

The present paper discusses the electromagnetohydrodynamic (EMHD) electroosmotic flow (EOF) and entropy generation of incompressible third-grade fluids in a parallel microchannel. Numerical solutions of the non-homogeneous partial differential equations of velocity and temperature are obtained by the Chebyshev spectral collocation method. The effects of non-Newtonian parameter *Λ*, Hartman number *Ha* and Brinkman number *Br* on the velocity, temperature, Nusselt number and entropy generation are analyzed in detail and shown graphically. The main results show that both temperature and Nusselt number decrease with the non-Newtonian physical parameter, while the local and total entropy generation rates exhibit an adverse trend, which means that non-Newtonian parameter can provoke the local entropy generation rate. In addition, we also find that the increase of non-Newtonian parameter can lead to the increase of the critical Hartman number *Hac*.

## 1. Introduction

Microfluidic devices are widely demonstrated in areas of biomedical and biochemical analysis, and have been one of the powerful tools for studying basic physical processes [1,2,3]. In these processes, pressure gradients, electrical fields, magnetic fields or their suitable combinations are the popular actuation mechanisms. Compared with the previous single pattern of pressure-driven flow, increasing attention has been attached to electroosmotic and electromagnetic actuation mechanisms in recent years. With the rapid development of lab-on-a-chip technologies, electroosmosis has been widely utilized with advantages of high reliability and simple operation [4]. A variety of theoretical and experimental studies of electroosmotic flow (EOF) in a microchannel have been extensively performed for both Newtonian fluid [5,6] and non-Newtonian fluid [7,8,9,10,11]. The hydrodynamic dispersion-combined electroosmosis and magnetohydrodynamic effect has been analyzed in microchannels with slowly varying wall zeta potentials [12]. Heat-transfer phenomena that are associated with electroosmotic and pressure-driven flows in microchannels have also been studied for thermally fully-developed flows [13,14,15] and thermally developing flows [16,17].

Aside from electroosmosis mentioned above, the magnetohydrodynamic (MHD) flow has also attracted much attention due to its prospective applications in engineering and medical fields [18,19,20]. Meanwhile, in order to achieve more efficient flow control, electromagnetohydrodynamic (EMHD) flow has also received widespread attention, i.e., both the external electrical and magnetic field are applied to the conducting fluid. The interaction of electric field and transverse magnetic field can produce the Lorentz force, which is a non-intrusive way to influence the motion of EMHD flow. Numerous theoretical and experimental works in the literature are available on the analysis of the behaviors of EMHD flow. Jang and Lee [21] found low magnetic field could bring about impressive increments to the fluid velocity. A practical EMHD pump has been constructed by Lemoff and Lee [22], in which an electrolytic solution was propelled by the Lorentz force along a micro-channel. Jian and Chang [23] obtained approximate analytical solutions of the EMHD velocity distribution under the influence of a non-uniform magnetic field. Under the combined action of electroosmotic and electromagnetic forces, the heat transfer characteristics of EMHD flows in a narrow channel have been analyzed by Chakraborty et al. [24]. Sarkar et al. [25] carried out a study on streaming potential of EMHD flow combined with interfacial slip through a microparallel channel, and the effects of electrical double-layer (EDL) formation were also taken into account. The results show that the flow rate was greatly improved, even at lower values of surface potential.

In recent years, it has been gradually realized that non-Newtonian fluids are more imperative than Newtonian fluids in a variety of industrial and engineering applications. For the non-Newtonian models, the relationship between shear stress and rate of strain is non-linear. Various non-Newtonian MHD flow models can be found in the existing literature [26,27,28,29] and hydrodynamic studies on non-Newtonian electroosmotic flows in reference [30,31]. Third-grade fluids model are able to discern normal stress differences and to describe shear thinning/thickening effect. Polymers, liquid metals, suspensions and so on belong to third-grade fluids. Wang and Jian [32] studied the EMHD third-grade fluids flow between two parallel microchannels and obtained the approximate analytical solutions of velocity and temperature by the perturbation method. Akgül et al. [33] discussed the analytical and numerical solutions of electroosmotical flow of third-grade fluid between parallel plates. Danish et al. [34] analyzed the flow characteristics of the Poiseuille and Couette–Poiseuille flow of third grade fluids through parallel plate. In addition, other relevant references on fluid motion and thermal transport of various fluids can be found, including third-grade fluids [35,36,37,38,39], Phan–Thien–Tan-nner (PTT) fluids [40,41,42], Maxwell fluids [43] and nanofluids [44,45].

Heat transfer processes are very common in nuclear reactor cooling, magnetic fluid power generation, and geophysical fluids, which are all accompanied by the thermodynamic irreversibility or entropy generation. Therefore, interest in the study of entropy generation has increased in the recent years. The possible causes of entropy generation are the heat transfer down temperature gradient, the influence of viscous dissipation, and the effect of convective heat transfer [46]. In the light of the close relationship between entropy generation and the lost available work (which is expressed by the Gouy Stodola theorem), more efficient thermal systems have been designed by decreasing the entropy generation. However, compared with the macro-scale process, the micro-fluidic entropy analysis is very limited in existing documents [47,48,49]. Pakdemirli and Yilbas [50] carried out entropy generation analysis of third-grade fluids model with Vogel viscosity in a pipe. The entropy generation rate for purely electroosmotic flows of NaCl electrolyte solution has been discussed in open-end and closed-end microchannels [51]. Jian [52] obtained the entropy generation rate of the MHD flow combined with electroosmotic effect in microparallel slit plates. Fersadou et al. [53] gave a numerical expression of entropy generation of MHD flow in a vertical porous channel. 

The spectral collocation method is a common numerical method for solving partial differential equations, especially in the field of computational fluid dynamics [54,55]. The main idea is to expand the solution of a partial differential equation into a finite series of smooth functions (usually orthogonal polynomials), and then, according to the original equation, to find the expansion coefficients. Hussaini [56] discussed the way to apply the spectral method in fluid mechanics, and in particular gave some details of numerical realization. The Chebyshev spectral method is one of the spectral methods to solve partial differential equations on a compact aperiodic interval, which use polynomial interpolation at Chebyshev points to estimate the solution of the equation.

Inspired by the above studies, the purpose of the present work is to investigate liquid flow features, heat-transfer characteristics and entropy generation rate of magnetohydrodynamic electroosmotic flow of third-grade fluids between two parallel plates. The momentum equations and energy equations are numerically solved by the Chebyshev spectral collocation method. Heat transfer characteristics, represented by the temperature distribution and Nusselt number, have been sketched in this study. Moreover, considering heat diffusion and heat convection, Joule heating effect, coupling electromagnetic diffusion, magnetic field and viscous friction, the entropy generation is discussed for different values of several pertinent non-dimensional parameters.

## 2. Formulation of the Problem

### 2.1. Physical Model and Explanation of the Problem

Here we consider thermally fully developed flow of third grade fluids in a microchannel under the magnetohydrodynamic electroosmotic influence. The flow is assumed to be steady, incompressible, viscous and electrically conducting. The physical model and the coordinate system are shown in Figure 1, Two external electrical field *E_x_* (in *x^*^*-axis direction) and *E_z_* (in *z*^*^-axis direction) are tangential to the charged surface, and a uniform magnetic field of strength *B* (in *y^*^*-axis direction) is perpendicular to the charged plates. A constant pressure gradient is imposed along the direction of flow, i.e., *x*^*^-axis direction. We assume the channel length *L* in *x*^*^-direction is much larger than both the channel height 2*h* in the *y*^*^-direction and the channel width *W* in the *z*^*^-direction, i.e., 2*h*, *W* << *L*; and the ratio of height 2*h* to width *W* of the channel is small enough, i.e., *δ* = 2*h*/*W* << 1. Therefore, the rectangular pipe flow is transformed into parallel plates flow. 

### 2.2. Electrical Potential Distribution 

The electric double layer (EDL) originates from the chemical contact interaction between the electrolyte solution and the channel wall. For a symmetric electrolyte solution, we assume that the ionic species concentration obeys the Boltzmann distribution. Therefore, the Poisson–Boltzmann equations are applied to describe the electric potential *ψ** of the EDL
(1)$$\nabla^{2}\psi^{*}=-\frac{\rho_{e}}{\varepsilon },$$
where *ρ*_e_ = −2*n*_0_*z_v_e*sinh(*z*_v_*e*_0_*ψ*/*k*_b_*T_a_*) is the local volumetric net charge density, *ε* is the dielectric constant of the medium, *n*_0_ is the bulk ionic concentration, *z_v_* is the ion valence, *e* is the electron charge, *k_b_* is the Boltzmann constant, and *T_a_* is the absolute temperature. The parallel plates are assumed to be charged and bear a uniform wall zeta potential of *ψ*_0_, which is small compared to the thermal potential, i.e., ($\left|z_{v}e_{0}\psi \right|<k_{b}T_{a}$). Due to Debye–Hückel linearization approximation (sinh(*z*_v_*e*_0_*ψ*/*k*_b_*T_a_*) ≈ *z*_v_*e*_0_*ψ*/*k*_b_*T_a_*), the Poisson–Boltzmann equation and corresponding boundary conditions are simplified as:(2)$$\frac{d^{2}\psi^{*}(y^{*})}{dy^{*}{}^{2}}=\kappa^{2}\psi^{*}(y^{*}),\;\kappa ={\left(\frac{2z_{v}^{2}e^{2}n_{0}}{\varepsilon k_{b}T_{a}}\right)}^{1/2},$$
(3)$$\psi^{*}|{}_{y^{*}=h}=\psi_{0},\;\frac{\partial \psi^{*}}{\partial y^{*}}|{}_{y^{*}=0}=0.$$
where *κ* is the Debye–Hückel parameter and 1/*κ* denotes the characteristic thickness of the EDL. The local volumetric net charge density *ρ*_e_ can be easily evaluated
(4)$$\rho_{e}=-\varepsilon \kappa^{2}\psi_{0}\cosh(\kappa y^{*})/\cosh(\kappa h).$$

### 2.3. Flow Analysis and Mathematical Formulation

The velocity of the incompressible flow is governed by the continuity equation and the momentum governing equation:(5)$$\nabla \cdot V^{*}=0$$
(6)$$\rho \frac{dV^{*}}{dt}=\nabla \cdot \tau +f$$
where *ρ* is the density of the fluid, ***V**** = (*u**, *v**, *w**) is the velocity vector, ***τ*** is stress tensor, and ***f*** is the body force vector acting on the flow, which is mainly composed of Lorentz force and the electrical force resulting from the electrokinetic effects. It can be written as
(7)$$f=\rho_{e}E+J\times B$$
where ***E*** = *E_x_**e**_x_* − *E_z_**e**_z_* is the applied electrical field vector along *x*^*^-axis and *z*^*^-axis directions, and both components in two directions are supposed to be invariant. ***B*** = *B**e**_y_* is the applied constant magnetic field along *y*^*^-direction. Here, we mention in particular that the induced magnetic field is neglected due to the small magnetic Reynolds number. ***J*** is the local ion current density vector and obeys the Ohm’s law:(8)$$J=\sigma (E+V^{*}\times B)$$
where *σ* is the electrical conductivity. In general, the Cauchy stress tensor ***τ*** for incompressible third grade fluids is given by [37]:(9)$$\tau =-pI+\mu A_{1}+\alpha_{1}A_{2}+\alpha_{2}A_{1}^{2}+\beta_{1}A_{3}+\beta_{2}(A_{1}A_{2}+A_{2}A_{1})+\beta_{3}(trA_{1}^{2})A_{1}{,}^{\;}$$
where *p* is the pressure, *I* is the identity tensor. *μ* denotes the dynamic viscosity and *α_i_* (*i* = 1, 2) and *β_i_* (*i* = 1, 2, 3) are the material constants. *Ai* (*i* = 1, 2, 3) are kinematic tensors with the following expressions:(10)$$A_{1}=(\mathrm{grad}V^{*})+{(\mathrm{grad}V^{*})}^{T},$$
(11)$$A_{n}=\frac{dA_{n-1}}{dt}+A_{n-1}(\mathrm{grad}V^{*})+{(\mathrm{grad}V^{*})}^{T}A_{n-1},n=2,\;3$$

Due to continuity Equation (5) and the hypotheses 2*h*, *W* << *L*, only the axial velocity component *u^*^*(*y*^*^) along *x*^*^-axis is taken into account. This assumption has been proven to correct by making a comparison with the analytical solution for velocity in three directions in reference [25].
(12)$$V^{*}=[u^{*}(y^{*}),0,0{]}^{\;}$$

Substituting the velocity field Equation (12) and Equation (7) in Equation (6), and the pressure gradient is assumed to be a constant *C*_0_, the momentum governing equation along *x*^*^-axis direction can be converted into the following form:(13)$$\mu \frac{d^{2}u^{*}}{dy^{*}{}^{2}}+2(\beta_{2}+\beta_{3})\frac{d}{dy^{*}}[(\frac{du^{*}}{dy^{*}}{)}^{3}]-\sigma B^{2}u^{*}+\sigma BE_{z}+\rho_{e}E_{x}=\frac{\partial P^{*}}{\partial x^{*}}=C_{0},$$

Equation (13) (See Brief Steps in Appendix A) is subjected to the following no-slip boundary conditions:(14)$$u^{*}(y^{*})=0,\;\mathrm{at}\;y^{*}=\pm h$$

To obtain the dimensionless form of Equation (13), non-dimensional parameters are defined as follows:(15)$$y=\frac{y^{*}}{h},\;u=\frac{u^{*}}{U_{eo}},\;U_{eo}=-\frac{\varepsilon \psi_{0}E_{x}}{\mu },\;\Lambda =\frac{(\beta_{2}+\beta_{3})U_{eo}{}^{2}}{\mu h^{2}},\;Ha=Bh\sqrt{\frac{\sigma }{\mu }},S=\frac{E_{z}h}{U_{eo}}\sqrt{\frac{\sigma }{\mu }},\Omega =\frac{h^{2}C_{0}}{\mu U_{eo}},K=\kappa H.$$
where *U_eo_* is the Helmholtz–Smoluchowski electroosmotic velocity, *Λ* is the dimensionless parameter related to the non-Newtonian behavior, *Ha* is the Hartman number, which represents the relative strength of the electromagnetic forces and the viscous forces, *S* is a non-dimensional quantity denoting the strength of the lateral electric field. *Ω* represents the estimate of applied pressure compared to electroosmotic force. *K* is so-called electrokinetic width.

So the dimensionless form of velocity in Equations (13) and (14) are:(16)$$\frac{d^{2}u}{dy^{2}}+6\Lambda {\left(\frac{du}{dy}\right)}^{2}\frac{d^{2}u}{dy^{2}}-Ha^{2}u+HaS+K^{2}\frac{\cosh(Ky)}{\cosh(K)}-\Omega =0.$$
(17)u(y)=0 ,at y=±1.

### 2.4. Thermal Transport for Thermally Fully Developed Flow

It is well known that viscous dissipation becomes significant in the microscale conduits. Sadeghi and Saidi concluded that viscous dissipation effects in combined pressure and electroosmotically driven flow had an important infection to the microscale thermal transport characteristics [57]. Considering the viscous dissipation, electromagnetic coupling heat and volumetric joule heating, the energy governing equation can be given as [15]:(18)$$\rho c_{p}\frac{dT^{*}}{dt}=k_{th}\nabla^{2}T^{*}+\tau :\mathrm{grad}(V^{*})+\frac{J\cdot J}{\sigma }$$
where *T^*^* is the local temperature of the liquid, *c_p_* is the specific heat at constant pressure, *k_th_* is the thermal conductivity of the liquid.

Considering a steady state and thermally fully developed flow, Equation (18) can be written as:(19)$$\begin{array}{l} \rho c_{p}u^{*}\frac{\partial T^{*}}{\partial x^{*}}=k_{th}(\frac{\partial^{2}T^{*}}{\partial x^{*}{}^{2}}+\frac{\partial^{2}T^{*}}{\partial y^{*}{}^{2}})+\mu {\left(\frac{du^{*}}{dy^{*}}\right)}^{2}+2(\beta_{2}+\beta_{3})\\ {\left(\frac{du^{*}}{dy^{*}}\right)}^{4}+\sigma (E_{x}{}^{2}+E_{z}{}^{2}+B^{2}u^{*}{}^{2}-2BE_{z}u^{*}) \end{array}$$

The second and forth terms on the right side of Equation (19) represent the volumetric energy generation caused by the viscous dissipation and Joule heat, which induced by Joule heating effect together with the contribution from electromagnetic effect, especially for the case of a large strength of magnetic field [58].

Furthermore, in a thermally fully developed case, we have:(20)$$\frac{\partial }{\partial x^{*}}\left[\frac{T_{s}{}^{*}(x^{*})-T^{*}(x^{*},y^{*})}{T_{s}{}^{*}(x^{*})-T_{m}{}^{*}(x^{*})}\right]=0.$$
where *T_s_*^*^ and *T_m_*^*^ are the local wall and mainstream temperature, respectively. Under the imposed constant heat flux (*q_s_*) boundary condition, i.e., *q_s_* = *ħ*(*T_s_*^*^ − *T_m_*^*^) = const., where *ħ* is the convective heat transfer coefficient, we obtain:(21)$$\frac{\partial T^{*}}{\partial x^{*}}=\frac{dT_{m}^{*}}{dx^{*}}=\frac{dT_{s}^{*}}{dx^{*}}=const\;\mathrm{and}\;\frac{\partial^{2}T^{*}}{\partial x^{*}{}^{2}}=0$$

Based on the above assumptions, the energy Equation (19) and relevant boundary conditions are:(22)$$\begin{array}{l} \rho c_{p}u^{*}\frac{dT_{m}^{*}}{dx^{*}}=k_{th}\frac{\partial^{2}T^{*}}{\partial y^{*}{}^{2}}+\mu {\left(\frac{du^{*}}{dy^{*}}\right)}^{2}+2(\beta_{2}+\beta_{3})\\ {\left(\frac{du^{*}}{dy^{*}}\right)}^{4}+\sigma (E_{x}{}^{2}+E_{z}{}^{2}+B^{2}u^{*}{}^{2}-2BE_{z}u^{*}) \end{array}$$
(23)$$q_{s}=k_{th}\frac{\partial T^{*}}{\partial y^{*}}{|}_{y^{*}=h},\;(\mathrm{or}\;T^{*}{|}_{y^{*}=\pm h}=T_{s}^{*}(x^{*}))\;\mathrm{and}\;\frac{\partial T^{*}}{\partial y^{*}}{|}_{y^{*}=0}=0.$$

An overall energy balance for an elemental control volume on a length of duct d*x^*^* was considered:(24)$$\begin{array}{l} 2\rho c_{p}hu_{m}^{*}dT_{m}^{*}=2q_{s}dx^{*}+2\sigma (E_{x}{}^{2}+E_{z}{}^{2})hdx^{*}\\ +\sigma \int_{-h}^{h}(B^{2}u^{*}{}^{2}-2E_{z}Bu^{*})dy^{*}dx^{*}+\mu \int_{-h}^{h}{\left(\frac{du^{*}}{dx^{*}{}^{}}\right)}^{2}dy^{*}dx^{*}. \end{array}$$
where *u_m_*^*^ denotes axial mean velocity, and can be written as:(25)$$u_{m}^{*}=\frac{1}{2h}\int_{-h}^{h}u^{*}dy^{*}.$$

Then, the constant mean temperature gradient d*T*^*^_*m*/_d*x*^*^ can be expressed from Equation (24) as:(26)$$\frac{dT_{m}^{*}}{dx^{*}}=\frac{1}{\rho c_{p}}M=const.$$
where,
(27)$$M=\frac{1}{2u_{m}^{*}h}\left(2q_{s}+2\sigma (E_{x}{}^{2}+E_{z}{}^{2})h+\sigma B^{2}A+\mu d\right)-2\sigma E_{z}B$$

The coefficients *A* and *D* of Equation (27) are given by:(28)$$A=\int_{-h}^{h}u^{*}{}^{2}dy^{*},\;d=\int_{-h}^{h}{\left(\frac{du^{*}}{dy^{*}}\right)}^{2}dy^{*}$$

It is convenient to express Equation (22) in the non-dimensional form by introducing the non- dimensional parameters and variables as follows:(29)$$\theta =\frac{T^{*}-T_{s}{}^{*}}{q_{s}h/k_{th}},\;Br=\frac{\mu \;U_{eo}^{2}}{q_{s}h},\;S_{x}=\frac{\sigma E_{x}^{2}h}{q_{s}},\;S_{z}=\frac{\sigma E_{z}^{2}h}{q_{s}}.$$

Physically, the parameters *S_x_* and *S_z_* stand for the relative strength of Joule heating to wall heat flux, which can be viewed as the dimensionless Joule heat parameters, and *Br* is Brinkman number, which represents the ratio of heat generated by viscous dissipation to the applied wall heat flux. Then the non-dimensional Equation (22) is expressed by:(30)$$\frac{\partial^{2}\theta }{\partial y^{2}}=-B_{r}{\left(\frac{du}{dy}\right)}^{2}-2\Lambda B_{r}{\left(\frac{du}{dy}\right)}^{4}+(2HaBrS+\frac{hMU_{eo}}{q_{s}})u-B_{r}Ha^{2}u^{2}-S_{x}-S_{z}$$

The corresponding boundary conditions of the dimensionless energy equation are:(31)$$\frac{\partial \theta }{\partial y}|{}_{y=0}=0,\;\frac{\partial \theta }{\partial y}|{}_{y=1}=1.$$

Based on the dimensionless variables defined earlier, the bulk mean temperature *θ_m_* can be defined as:(32)$$\theta_{m}=\frac{\int_{-1}^{1}u\theta dy}{\int_{-1}^{1}udy}=k_{th}\frac{T_{m}{}^{*}-T_{s}{}^{*}}{q_{s}h}.$$

An important heat transfer parameter expressed as Nusselt number *Nu* can be written as:(33)$$Nu=\frac{\hbar d_{h}}{k_{th}}=\frac{q_{s}d_{h}}{k_{th}(T_{s}^{*}-T_{m}^{*})}.$$
where *D_h_* denotes hydrodynamic diameter and *D_h_* = *h* for a half of microchannel height. From Equations (32) and (33), the finial local Nusselt number (at the upper wall) can be expressed as:(34)$$Nu=-\frac{1}{\theta_{m}}.$$

### 2.5. Entropy Generation Rate

According to the entropy generation minimization concept [46,59], the local volumetric rate of entropy generation based on the above obtained velocity and temperature field can be expressed as:(35)$$S_{G}^{*}=S_{G,H}^{*}+S_{G,J}^{*}+S_{G,C}^{*}+S_{G,M}^{*}+S_{G,V}^{*}.$$
where *S_G_*^*^ is the volumetric entropy generation rate, which consists of five parts: heat diffusion irreversibility, Joule heating effect, coupling electromagnetic diffusion, magnetic field and viscous friction of the fluids. In this work, they are written respectively as:(36)$$S_{G,H}^{*}=\frac{k_{th}}{T^{*}{}^{2}}\left[{\left(\frac{\partial T^{*}}{\partial x^{*}}\right)}^{2}+{\left(\frac{\partial T^{*}}{\partial y^{*}}\right)}^{2}\right],\;S_{G,J}^{*}=\frac{\sigma (E_{x}{}^{2}+E_{z}{}^{2})}{\left|T^{*}\right|},\;S_{G,C}^{*}=\frac{2\sigma E_{z}Bu^{*}}{\left|T^{*}\right|},S_{G,M}^{*}=\frac{\sigma B^{2}u^{*}{}^{2}}{\left|T^{*}\right|},\;S_{G,V}^{*}=\frac{\mu }{\left|T^{*}\right|}{\left(\frac{du^{*}}{\partial y^{*}}\right)}^{2}$$

By use of the characteristic entropy transfer rate (*k_th_/h*^2^), the dimensionless form of entropy generation rate can be given as:(37)$$S_{G}^{}=S_{H}+S_{J}^{}+S_{C}^{}+S_{M}^{}+S_{V}^{}$$
where,
(38)$$S_{H}^{}=\frac{1}{{\left(\theta +\Theta \right)}^{2}}\left[{\left(\frac{\partial \theta }{\partial y}\right)}^{2}+\frac{F^{2}}{Pe^{2}}\right],\;S_{J}^{}=\frac{1}{\left|\theta +\Theta \right|}\left(S_{x}+S_{z}\right),\;S_{C}^{}=\frac{2SHaBru}{\left|\theta +\Theta \right|},\;S_{M}^{}=\frac{BrHa^{2}u^{2}}{\left|\theta +\Theta \right|},\;S_{V}^{}=\frac{Br}{\left|\theta +\Theta \right|}{\left(\frac{du}{dy}\right)}^{2}.$$
where *Θ* = (*k_th_T_s_*^*^)/(*q_s_h*) is a constant determined by the unaltered wall temperature and the heat flux, *Pe* = (*ρ**c_p_U_eo_h*)/*k_th_* is Peclet number, and the variable *F* in Equation (38) is:$$F=\frac{1+S_{x}+S_{z}+BrHa^{2}I_{2}-2BrHaSu_{m}+BrI_{1}}{u_{m}}$$
where $u_{m}=\frac{1}{2}\int_{-1}^{1}udy$, $I_{1}=\int_{-1}^{1}{\left(\frac{du}{dy}\right)}^{2}dy$, $I_{2}=\int_{-1}^{1}u^{2}dy$.

In addition, the total non-dimensional entropy generation can also be obtained:(39)$$S_{total}=\int_{-1}^{1}\;S_{G}\;dy.$$

## 3. Numerical Solution

Among numerous numerical methods, the Chebyshev spectral method has higher accuracy and wider application. In this paper, by utilizing the Chebyshev spectral collocation method, we study the non-dimensional EMHD velocity, temperature, Nusselt number and entropy generation of third-grade fluids between two parallel micro-plates, owing to the fact that the analytical solutions of these physical quantities are difficult to obtain for third-grade fluids when EMHD electroosmotic effects are all taken into condition.

The physical domain is [−1, 1] in the present analysis, and the Chebyshev points *y_j_* = cos(*j*π/*N*), *j* = 0, 1, …, *N* are chosen to discretize the interval. Let *u* = [*u*(*y*_0_), *u*(*y*_1_), …, *u*(*y_N_*)] be the undetermined vector at the Chebyshev points, then we obtain a Chebyshev polynomial *P* of degree at most equal to *N*, i.e., *P*(*y_i_*) = *u*(*y_i_*), *i* = 0, 1, …, *N*. By differentiating *P* and evaluating at the grid points, we can transform the differential equation into linear algebraic equations, and the numerical solution of Equation (16) under Equation (17) can be obtained. By use of the velocity values obtained, the value of dimensionless temperature *θ* can be easily computed at each Chebyshev point by Equation (30) which is a second order differential equation, and then the value of the Nusselt number and entropy generation can be easily calculated by Equations (34) and (37).

## 4. Results and Discussion

In the following sections, the distributions for dimensionless velocity, temperature, Nusselt number and entropy generation rate will be discussed. The effects of non-Newtonian parameter, Joule heating, magnetic field intensity and electro-kinetic parameters on the above physical quantities will be shown graphically. Before proceeding, the permissible ranges of relevant physical parameters should be given firstly.

For typical microscale fluid flow, half-height of the channel *h* is about 100 μm, viscosity *μ*~10^−3^ kg/(ms), electrical conductivity *σ*~2.2 × 10^−4^–10^6^ S/m. The applied magnetic field *B* is 0.018–0.44T [21], so the range of Hartmann number *Ha* varies from 0 to 3 calculated from Equation (15) [32,60]. The strength of the applied electric field varies from 0 to 20 V/m and the electroosmotic velocity is *U*_eo_ ~100 μm/s. Generally, *S_x_* and *S_z_* are both positive, and for simplicity we assume *S_x_* + *S_z_* = 1 [51]. We suppose the electrokinetic width *K* = 10 unless there is a special announcement [43], which is a typical thickness value for non-overlapping EDLs. Brinkman number (*Br*) is 0–0.04. What is worth noting, according to the theoretical analysis given by Sarkar et al. [25], is that the value of lateral electric field should not be too large, otherwise the induced transverse flow will not be neglected, which will contradict the assumption of unidirectional flow. Thus, if there is no special statement, the value of *S* is set to 1. In addition, Péclet number *Pe* is restricted to 0.5 [61].

Firstly, we have conducted comparisons including two special cases. In Figure 2, the result of the present numerical velocity (for *Λ* = 0) is compared to the result obtained by Chakraborty et al. [24], who have discussed the Newtonian fluid which is subjected to the combined action of electroosmotic and electromagnetic forces. Secondly, in Figure 3 the present results of velocity distributions and temperature distributions (for *S* = 0 and *Ha* ≈ 0) for different non-Newtonian parameters *Λ* are compared to the results of Akgül et al. [33], who have obtained approximate analytical solutions of the electroosmotic flow of the third grade fluids by the perturbation techniques. We can see that the present results are in good agreement with the earlier conclusion when the non-Newtonian parameter *Λ* is small enough, however, for larger *Λ* (0.025), perturbation method is no longer appropriate to solve the present problem. Therefore, the spectral collocation method in our analysis has a broader application.

### 4.1. Velocity Analysis

In order to highlight the electromagnetohydrodynamic electroosmotic effects, in the following sections we will discuss the flow and the heat-transfer characteristics without regard for the pressure gradient effect. Figure 4 delineates the profiles of flow velocity with different fluid physical parameters. It can be seen from Figure 4a that the velocity decreases with the increase of non-Newtonian parameter *Λ*. The reason is that the increase of *Λ* is actually related to the increase of the viscosity of third-grade fluids. The effects of Hartman number *Ha* on the dimensionless velocity are shown in Figure 4b,c. The results are the same as those discussed in reference 24, for small *Ha* (*Ha* ≤ 1), the aiding force (*σBE*_z_) is greater than the retarding one (−*σB*^2^*u*) which can be seen in Equation (16), therefore, the velocity increases with *Ha* (as depicted in Figure 4b). With the further increase of *Ha,* the equilibrium between aiding force and retarding force arrives, the corresponding value of *Ha* is so-called critical *Hac*. For *Ha* beyond the critical *Hac* (as depicted in Figure 4c), the retarding magnetic force becomes the leading factor and triggers a progressive reduction in the flow velocity. Finally, we can conclude from Figure 4d that the velocity increases with the augment of the strength of the lateral electric field *S*. Clearly by increasing the magnitude of the lateral electric field *S*, the aiding force (*HaS*) is dominant which promotes the increase of velocity. In order to gain a better understanding of the critical *Hac*, the profiles of average velocity have been pictorially depicted for *S* = 1 in Figure 5. It is easy to observe that the average velocity profile has been divided into two regions by the critical *Hac*, and shows an increasing-decreasing trend with the increase of *Ha*. It is worth mentioning that the value of critical *Hac* shows an increasing trend with the increase of non-Newtonian parameter *Λ*.

### 4.2. Temperature Analysis

The effects of various dimensionless parameters, including non-Newtonian parameter *Λ*, the magnetic field *Ha* and viscous dissipation *Br*, on the distribution of fluid temperature are discussed in Figure 6. Firstly, a decreasing trend in temperature with non-Newtonian parameter *Λ* is observed in Figure 6a. This variation tendency is consistent with the previous result in velocity because the decrease of velocity leads to the decrease of heat exchange in non-Newtonian fluids. From Figure 6b,c, it can be found that the magnitude of dimensionless temperature increases with small *Ha* (*Ha* < *Hac*), the variation tendency is opposite for big *Ha* (*Ha* > *Hac*), and the maximum temperature has been obtained in the center of channel. Finally, the effects of viscous dissipation *Br* are discussed in Figure 6d and we can see that the increase of *Br* can trigger a tiny increase in temperature. The reason is that the viscous dissipation can be viewed as an energy source to increase the temperature of the fluid.

To further understand the heat transfer characteristics, the variations of Nusselt number *Nu* with Brinkman number *Br* for the different values of Hartmann number *Ha* and non-Newtonian parameter *Λ* have been depicted in Figure 7. First it can be seen that, no matter what value *Ha* takes, the augment of *Br* results in continuous reduction of the Nusselt number *Nu*. The reason lies in the magnitude of the quantity *T_s_*^*^−*T_m_*^*^ gradually increasing with the Brinkman number. The augment of values of *Ha* leads to a decreasing-increasing variation trend of *Nu*, regardless of the magnitude of Brinkman number. In Figure 7a, in the case of small *Ha* (i.e., the aiding effect of lateral electric field is stronger than that of magnetic field), the increase of flow velocity with *Ha* results in the decrease of the bulk mean temperature and the convective heat-transfer coefficient, and then the Nusselt number presents a downward trend. For different non-Newtonian parameter *Λ*, the Nusselt number *Nu* profiles have been graphically depicted in Figure 7c. We can observe that the Nusselt number decreases with increasing non-Newtonian parameter *Λ* for the case of *Ha* = 3, which is a natural result corresponding to the previous variations in velocity and temperature.

### 4.3. Entropy Generation Analysis

The profiles of local entropy generation of third-grade fluids at different values of *Ha* have been delineated in Figure 8a,b. The variation trend of local entropy generation is opposite for low values of *Ha* and the high ones. For low values of Hartman number, we can reduce entropy by increasing the intensity of the magnetic field, and we can see the maximum values of entropy have been reached at the center of parallel plates. This is a reasonable result by noticing the fact that most changes of velocity and temperature distributions in the previous figures occur at the center.

Moreover, the influence of *Br* on the local entropy generation is illustrated in Figure 8c. It can be observed that the local entropy generation shows an increasing trend with the increase of *Br*. From Figure 8d, it can be seen that entropy reduction can be achieved by reducing the value of non-Newtonian parameter *Λ*. Finally, Figure 8e shows the entropy generation number falls with increasing Péclet number *Pe*. This is physically true due to the augment of *Pe* meaning a decrease of thermal conductivity of the fluid which can be seen in the definition of *Pe* and, therefore, a decreasing trend in entropy generation is observed [62].

In Figure 9, we can observe that the total entropy generation rates both show an increasing trend with the increase of *Λ* and *Br* for any fixed value of *Ha*, similar to the tendency of the local entropy generation *S_G_*. In particular, it can be noticed that the total entropy generation rate presents a decreasing-increasing variation trend with *Ha*, which is consistent to the former conclusion in Figure 8a,b.

Figure 10 illustrates total entropy generation rate versus magnetic field parameter *Ha* and the ratio of the viscous dissipation to the applied wall heat flux *Br* for different values of the lateral electric field *S*. There is a great decrease in the total entropy generation rate with the increasing value of *S* for given values of *Ha* and *Br*.

## 5. Conclusions

In this study, we have discussed a mathematical model for describing the electromagnetohydrodynamic flow and entropy generation of third-grade fluids between two parallel microplates combined with electroosmotic effects. The Chebyshev spectral collocation method has been applied to obtain the numerical solutions of the dimensionless velocity, temperature and entropy generation rate under the unidirectional flow assumption. The influences of dimensionless governing parameters, including non-Newtonian parameter (*Λ*), magnetic field (*Ha*) and viscous dissipation (*Br*) on the above obtained physical quantities are systematically investigated. The following conclusions can be drawn from the above theoretical analysis and numerical simulation. First of all, the results show that effect of non-Newtonian parameters are significant on fluid velocity, temperature and entropy generation rate. The dimensionless flow velocity and temperature are observed to decrease with the increase of non-Newtonian parameters and their maximum values have been reached when *Λ* = 0 (i.e., the fluid is Newtonian fluid). Under the combined action of electrical field and magnetic field, the variation profiles of velocity and temperature with increasing *Ha* has been divided into two regions, separated by the critical Hartmann number *Hac*, which also increases with non-Newtonian parameter. In addition, we observe that the Nusselt number shows decreasing behavior for an increasing non-Newtonian parameter *Λ.* Finally, we find that non-Newtonian characteristic can stimulate both the local and the total entropy generation rate.

## Figures and Tables

**Figure 1 micromachines-11-00418-f001:**
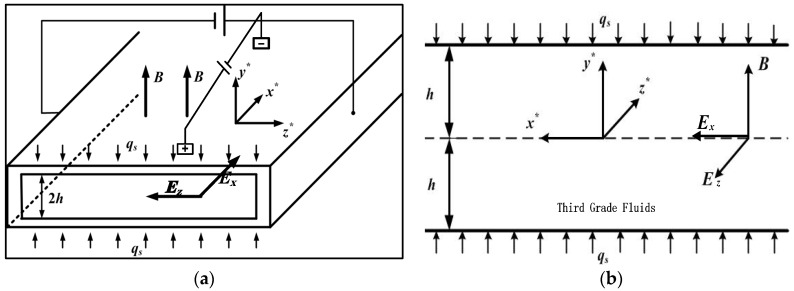
Schematic diagram of the physical model. (**a**) Three-dimensional (3D) view of the electromagnetohydrodynamic (EMHD) micro-pump; (**b**) Duct’s cross section of the EMHD micro-pump.

**Figure 2 micromachines-11-00418-f002:**
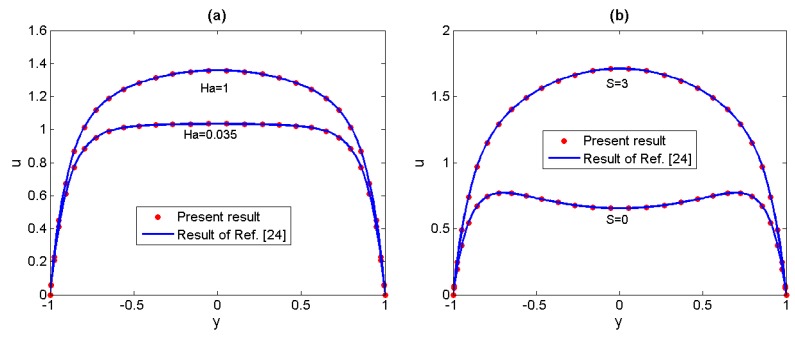
Comparisons of numerical velocity in our analysis with the analytical solutions of Chakraborty [24] for Newtonian fluid. (**a**) *Λ* = 0, *S* = 2, *Ω* = 0, *K* = 10; (**b**) *Λ* = 0, *Ha* = 1, *Ω* = 0, *K* = 10.

**Figure 3 micromachines-11-00418-f003:**
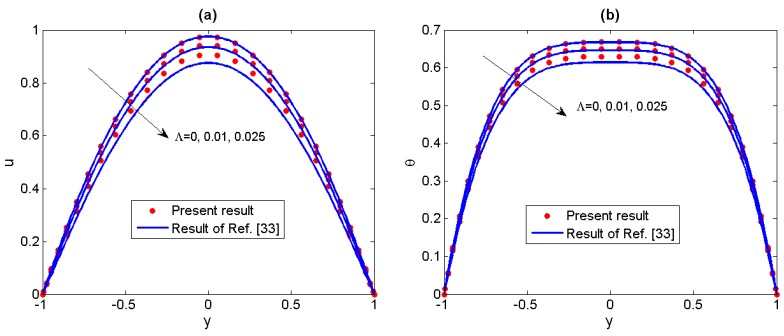
Comparisons of numerical velocity (**a**) and temperature (**b**) in our analysis with the approximate analytical solutions of Akgül [33] for different non-Newtonian parameters *Λ* (*S* = 0, *Ha* ≈ 0, *Ω* = −2, *K* = 10, *Br* = 2).

**Figure 4 micromachines-11-00418-f004:**
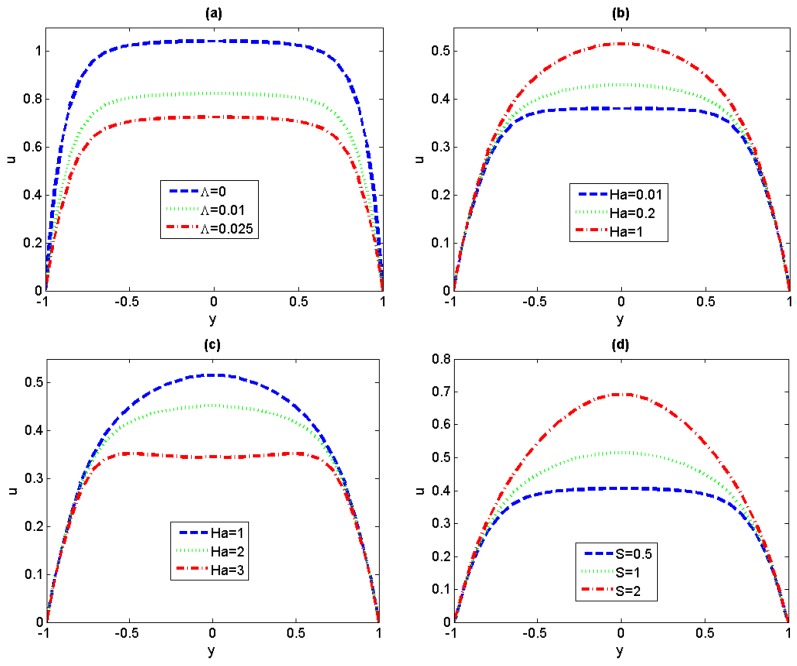
Variations of flow velocity for different fluid physical parameters. (**a**) *S* = 1, *Ω* = 0, *Ha* = 0.1; (**b**,**c**) *Λ* = 0.5, *S* = 1, *Ω* = 0; (**d**) *Λ* = 0.5, *Ha* = 1, *Ω* = 0.

**Figure 5 micromachines-11-00418-f005:**
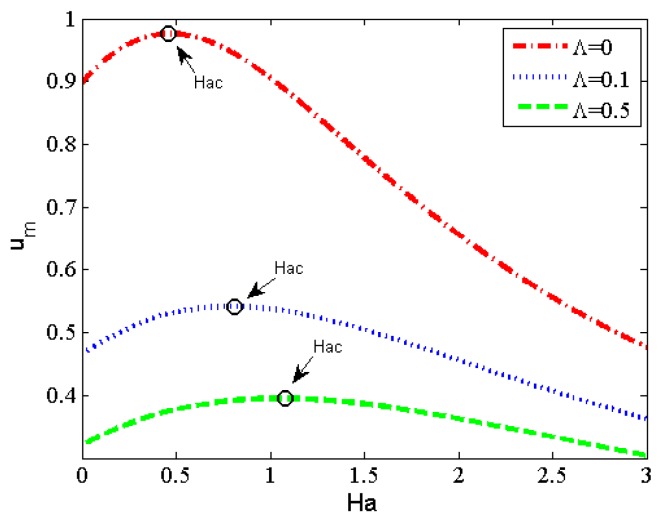
Variations of the average velocity with *Ha* for different magnitudes of *Λ* (*S* = 1).

**Figure 6 micromachines-11-00418-f006:**
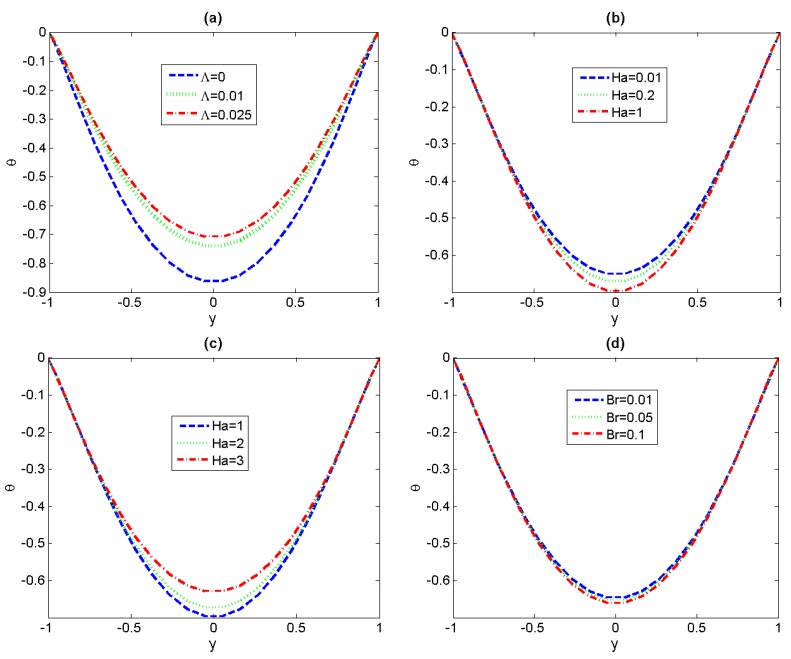
Variations of temperature for different fluid physical parameters. (**a**) *S* = 1, *Ω* = 0, *Ha* = 0.1, *Br* = 0.1; (**b**,**c**) *Λ* = 0.5, *S* = 1, *Ω* = 0, *Br* = 0.1; (**d**) *Λ* = 0.5, *Ha* = 1, *Ω* = 0, *S* = 1.

**Figure 7 micromachines-11-00418-f007:**
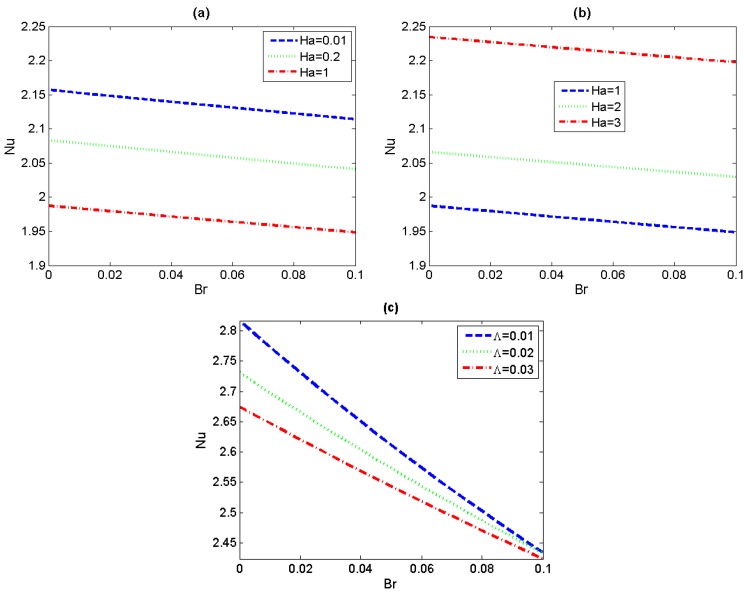
Variations of *Nu* versus *Br* with *Ha* and *Λ* (*S* = 1, *Ω* = 0), (**a**,**b**) *Λ* = 0.5; (**c**) *Ha* = 3.

**Figure 8 micromachines-11-00418-f008:**
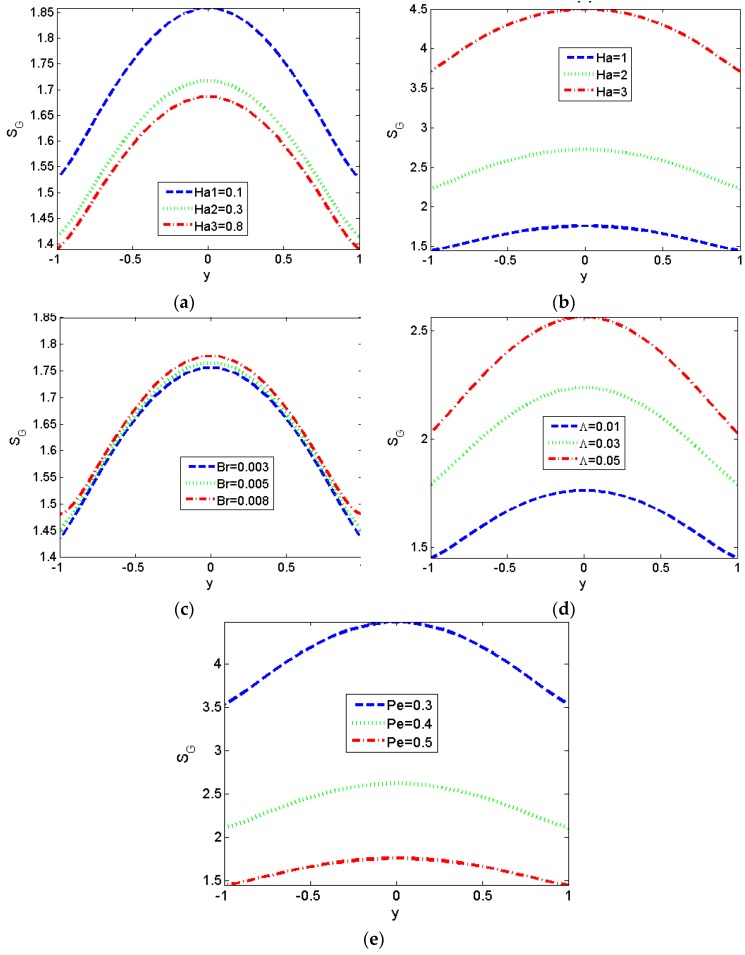
Variations of local entropy generation *S_G_* for different values of *Ha*, *Br* and *Λ*. (*S* = 1). (**a**,**b**) *Λ* = 0.01, *Br* = 0.005, *Pe* = 0.5; (**c**) *Λ* = 0.01, *Ha* = 1, *Pe* = 0.5; (**d**) *Br* = 0.005, *Ha* = 1, *Pe* = 0.5; (**e**) *Λ* = 0.01, *Ha* = 1, *Br* = 0.005.

**Figure 9 micromachines-11-00418-f009:**
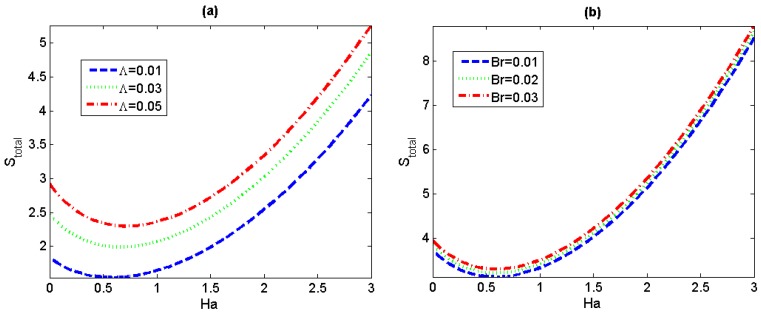
Variations of total entropy generation S_total_ for different value of *Λ* and *Br*. (*S* = 1) (**a**) *Br* = 0.005, *Pe* = 0.5; (**b**) *Λ* = 0.01, *Pe* = 0.5.

**Figure 10 micromachines-11-00418-f010:**
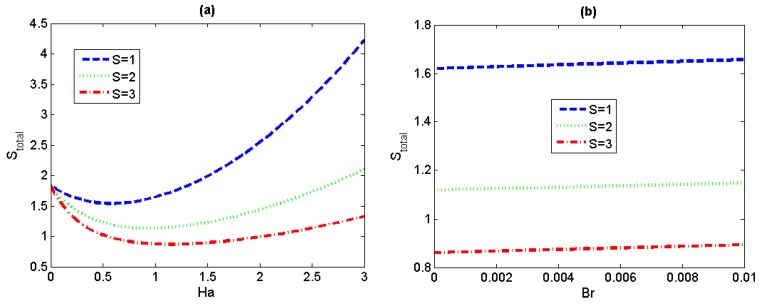
Variations of total entropy generation *S*
_total_ versus *Ha* and *Br* for different value of *S* (*Λ* = 0.01, *Pe* = 0.5). (**a**) *Br* = 0.005; (**b**) *Ha* = 1.

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
