# Peer review of "Electromagnetohydrodynamic Electroosmotic Flow and Entropy Generation of Third-Grade Fluids in a Parallel Microchannel"

_micromachines, 2020, doi:10.3390/mi11040418_

Round 1

Reviewer 1 Report

Can be accepted with minor revision. Authors need to clarify if this work is developed from published articles such as 

10.1016/j.ijheatmasstransfer.2018.06.147

10.1016/j.colsurfa.2016.01.006

 https://doi.org/10.1017/jmech.2016.57

Reviewer 2 Report

The authors describe the  effect of electromagnetohydrodynamic   (EMHD), electroosmotic  flow  (EOF)  and  entropy  generation  of  incompressible  third  grade  fluids  in  a  parallel    microchannel.   They propose some numerical   solutions   of   the   non-­‐homogeneous   partial   differential   equations   of  velocity   and   temperature.   The   effects   of   some physical parameters like  Hartman   number, Brinkman   number, temperature,   Nusselt   number   and   entropy   generation   are analyzed and   shown  graphically.

In my opinion the article is addressed to a restricted board of interest. Only scientists of the field could easily read the description and the information reported.

They skipped through some important points, like the accurate description of the influence of controlling entropy for the microfulidic devices. 

I would suggest to modify and clarify the description reported in page 2 from line 69 to 81.

During the description of the work, the authors did a lot of assumption that are not immediate for the general reader, i.e. page 8 line 262 they reported the value of fixed parameters without giving more information about that.

Some symbols are not visible in the pdf file (page 11 line 331) and more in general the article is difficult to read.

Probably I would encourage the submission to a more theoretical focused journal.

Reviewer 3 Report

The authors numerically studied the EMHD electroosmotic flow. The governing equations were solved by Chebyshev spectral collocation method. The effects of various parameters on flow velocity and entropy generation were discussed. The paper is well written and organized. I suggest the publication after minor changes.

(1) It is better to include some introduction about the Chebyshev Spectral Collocation method in Section 1.

(2)In fig. 3, the large difference is found for ∧=0. Can authors comment on this?

Round 2

Reviewer 1 Report

Can be accepted in present form

Reviewer 2 Report

Thanks for addressing the comments